# Unsupervised Pretraining for Fact Verification by Language Model Distillation

**Adrián Bazaga, Pietro Liò & Gos Micklem**
University of Cambridge
`{ar989,pl219,gm263}@cam.ac.uk`

## Abstract

Fact verification aims to verify a claim using evidence from a trustworthy knowledge base. To address this challenge, algorithms must produce features for every claim that are both semantically meaningful, and compact enough to find a semantic alignment with the source information. In contrast to previous work, which tackled the alignment problem by learning over annotated corpora of claims and their corresponding labels, we propose SFAVEL (*S*elf-supervised *Fa*ct *Ve*rification via *L*anguage Model Distillation), a novel unsupervised pretraining framework that leverages pre-trained language models to distil self-supervised features into high-quality claim-fact alignments without the need for annotations. This is enabled by a novel contrastive loss function that encourages features to attain high-quality claim and evidence alignments whilst preserving the semantic relationships across the corpora. Notably, we present results that achieve a new state-of-the-art on FB15k-237 (+5.3% Hits@1) and FEVER (+8% accuracy) with linear evaluation.

## 1 Introduction

In recent years, the issue of automated fact verification has gained considerable attention as the volume of potentially misleading and false claims rises (Guo et al., 2022), resulting in the development of fully automated methods for fact checking (see Thorne et al. (2018); Zubiaga et al. (2018); Guo et al. (2022); Vladika & Matthes (2023); Das et al. (2023) for recent surveys). Pioneering research in the field of Natural Language Processing (NLP) has led to the emergence of (large) language models (LMs) (e.g. Raffel et al. (2020b); Brown et al. (2020); Radford et al. (2019; 2018)). These models have been successful in many applications due to the vast implicit knowledge contained within them, and their strong capabilities for semantic understanding of language. However, issues around fact hallucination have gained considerable attention (Huang et al., 2023; Liu et al., 2023) and are a major concern in the widespread usage of LLM-based applications across different settings.

As the world becomes more aware of the issues around information trustworthiness, the importance of developing robust fact verification techniques grows ever more critical. Historically, the design of fact verification methods has been enabled by the creation of annotated datasets, such as FEVER (Thorne et al., 2018) or MultiFC (Augenstein et al., 2019), of appropriate scale, quality, and complexity in order to develop and evaluate models for fact checking. Most recent methods for this task have been dominated by two approaches: natural language inference (NLI) models (e.g., Si et al. (2021); Zhu et al. (2021); Thorne et al. (2018); Luken et al. (2018); Yin & Roth (2018); Ye et al. (2020)), and knowledge graph-augmented methods (e.g. Zhou et al. (2019); Zhong et al. (2020); Chen et al. (2021a;b); Liu et al. (2021)). These proposals mainly leverage the NLI methods to model the semantic relationship between claim and evidence, or further make use of the knowledge graph (KG) structure to capture the features underlying between multiple pieces of evidence.

However, these studies have largely relied on annotated data for model training, and while gathering data is often not difficult, its labeling or annotation is always time-consuming and costly. Thus, an emerging trend in the literature (Chen et al., 2020a;c; Caron et al., 2020; He et al., 2020) is to move away from annotation-dependant methods and try to learn patterns in the data using unsupervised training methods. With the advent of unsupervised training methods, new avenues have opened for research into leveraging the huge amounts of unlabeled data to achieve better performance more efficiently. Despite significant advancements in the field of unsupervised learning, only a handful

of strategies have been proposed for textual fact verification (e.g. Jobanputra (2019); Kim & Choi (2020); Jolly et al. (2022); Zeng & Gao (2023)). Thus there are still opportunities for the development of unsupervised techniques tailored specifically for such tasks.

Following recent trends in unsupervised methods, we eliminate data annotation requirements and instead, without human supervision, automatically try to identify relevant evidence for fact-checking. Thus, in this paper we present SFAVEL (Self-supervised Fact Verification via Language Model Distillation), which introduces a novel self-supervised feature representation learning strategy with well-designed sub-tasks for automatic claim-fact matching. SFAVEL leverages pre-trained features from Language Models and focus on distilling them into compact and discrete structures that attain a high alignment between the textual claims to be verified, and their corresponding evidence in the knowledge graph. In particular, our contributions are summarized as follows:

- We introduce Self-supervised Fact Verification via Language Model Distillation (SFAVEL), a novel unsupervised pretraining method tailored for fact verification on textual claims and knowledge graph-based evidence by language model distillation.
- We demonstrate that SFAVEL achieves state of the art performance on the FEVER fact verification challenge and the FB15k-237 dataset when compared to both previous supervised and unsupervised approaches.
- We justify SFAVEL's design decisions with ablation studies on the main architectural components.

## 2    RELATED WORK

**Fact verification with pre-trained language models**    Most recent works typically divide the fact verification task into two stages. The first stage retrieves a relatively small subset of evidence from a knowledge source (e.g. a knowledge graph) that is relevant to verify a given claim. The second stage performs reasoning over the retrieved evidence to discern the veracity of the claim. Such retrieval-and-reasoning approaches aim to reduce the search space, and have proven their superiority over directly reasoning on the whole knowledge graph (Chen et al., 2019; Saxena et al., 2020). In order to match evidence with claims, a typical approach is to devise claim-fact similarities using semantic matching with neural networks. Due to the great semantic understanding recently demonstrated by pre-trained language models (PLMs), some recent works employ PLMs for addressing the claim-fact semantic matching task. In this vein, some works exploit the implicit knowledge stored within LMs for performing zero-shot fact checking, without any external knowledge or explicit evidence retrieval (Lee et al., 2020; Yu et al., 2023). However, such methods are prone to suffer from hallucination errors, resulting in incorrect predictions. Other work, such as ReAct (Yao et al., 2022), explores the use of LMs to generate both reasoning traces and task-specific actions over a knowledge base by in-context learning via prompting, overcoming prevalent hallucination issues by interacting with a Wikipedia API. However such an approach is limited by input length making it impractical in complex tasks. In SFAVEL, we distill the features of recent pre-trained language models to yield highly-correlated claim and evidence embeddings. We make use of a set of eight language models as backbones because of their quality, but note that SFAVEL can work with any language model features.

**Unsupervised pre-training methods for fact verification**    Learning meaningful features for claim-fact matching without human labels is a nascent research direction in fact verification approaches, with recent works relying on self-supervised techniques. For instance, CosG (Chen et al., 2021d) proposes a graph contrastive learning approach to learn distinctive representations for semantically similar claims with differing labels, with the goal of mitigating the over-smoothing issues commonly found in graph-based approaches. The model incorporates both unsupervised and supervised contrastive learning tasks to train a graph convolutional encoder, enhancing the representation of claim-fact pairs in the embedding space. Mu et al. (2023) presents SSDL, a multi-task learning strategy that initially builds a student classifier using both self-supervised and semi-supervised methods, and then fine-tunes the classifier using distilled guidance from a larger teacher network that remains frozen during training. However, this method requires pairs of text claims and corresponding visual information as training data. Chen et al. (2021c) introduces KEGA, a knowledge-enhanced graph attention network for fact verification, which uses external knowledge bases to improve claim and

evidence representations. It uses a contrastive learning loss to capture graph structure features. With BERT as its backbone, the model includes knowledge from WordNet and pre-trained knowledge base embeddings to enrich token representations, while a graph attention network (Veličković et al., 2018) and the contrastive loss further enhance the model's ability to reason. LaPraDoR (Xu et al., 2022) introduces a pre-trained dense retriever approach with contrastive learning for unsupervised training of query and document encoders. The method is applied in a variety of text retrieval challenges, with FEVER being one of them. However, it shows a significant performance gap when compared against the supervised state-of-the-art approaches for FEVER. One of the main reasons is the lack of task-specific contrastive functions. In contrast to these works, SFAVEL is designed with a task-specific self-supervised feature representation strategy, leveraging language model distillation to achieve high-quality claim-fact unsupervised matching on large scale datasets for fact verification.

**Knowledge distillation**   Knowledge distillation seeks to transfer the knowledge from a (usually large) model, called the teacher, to another (usually small) model, called the student. This technique is often used for increasing the performance of the small model. One of the first approaches for knowledge distillation was proposed by Hinton et al. (2015), via minimizing the KL-divergence between the teacher and student's logits, using the predicted class probabilities from the teacher as soft labels to guide the student model. Instead of imitating the teacher's logits, Romero et al. (2015) distilled knowledge by minimizing the $\mathbb{L}_2$ distance between the intermediate outputs of the student and teacher. Park et al. (2019) aligned the pair-wise similarity graph of the student with the teacher, and Zagoruyko & Komodakis (2017) used the attention map generated by the teacher to force the student to attend to the same areas as the teacher. More recently, knowledge distillation has been extended for self-supervised settings. For instance, Tian et al. (2022) use the contrastive loss to enforce cross-modality consistency. Xu et al. (2020) and Fang et al. (2021) aim at aligning the features between views of the same instances by computing pair-wise similarities between the student's outputs and features kept in a feature memory bank produced by the teacher. In this work, we propose to transfer the semantic knowledge of a pre-trained language model within a new type of self-supervised task, fact verification, by leveraging the language model capabilities of language understanding to guide the student to produce high-quality features for claim-fact matching.

## 3 OVERVIEW OF THE APPROACH

In this section we present our proposed unsupervised approach, SFAVEL, in detail as illustrated in Figure 1a. First, we begin with the data processing pipeline. Then, we detail our proposed pre-training methodology, followed by details of the different components of our proposed contrastive loss function. Finally, we describe the optional adaptation phase, where we fine-tune the model pre-trained with our framework for a downstream fact verification classification task.

### 3.1 DATA PROCESSING PIPELINE

Throughout this section, we assume to have sampled a batch of unlabeled claims $x = \{x_i\}_{i=1}^N$, with $x_i$ being the $i^{th}$ claim and $N$ the batch size. In addition, we assume access to a knowledge base represented as a knowledge graph $G(\varepsilon, \mathcal{R})$, where $\varepsilon, \mathcal{R}$ are the set of real-world entities associated with a name (e.g. Barack Obama, New York City), and the relationship type between entities (e.g. $was\ born\ in$), respectively. We also define $F$ as the set of facts contained in $G$, represented as triples of subject, relation, object, named as head, relation and tail. Then, $G$ can be defined as $G = \{(h_i, r_i, t_i) \mid h_i, t_i \in \varepsilon, r_i \in \mathcal{R}\}$, where $i \in \{1, \ldots, |F|\}$.

### 3.2 PRETRAINING METHOD

As shown in Figure 1a, our pre-training approach uses a pre-trained language model to obtain a feature tensor for each of the input claims. In SFAVEL, we use a knowledge model to embed facts from the knowledge graph, and a scoring module that scores each of such facts conditioned upon a specific claim. To tackle the discrepancy in information-level between the features from the pre-trained language model and the knowledge model, we introduce an unsupervised distillation loss that encourages the representation of the claim and its related knowledge facts to be mapped close together in the feature space of the knowledge model. A scoring loss function encourages the scoring module to provide higher scores for positive facts than to randomly-generated negative facts. To

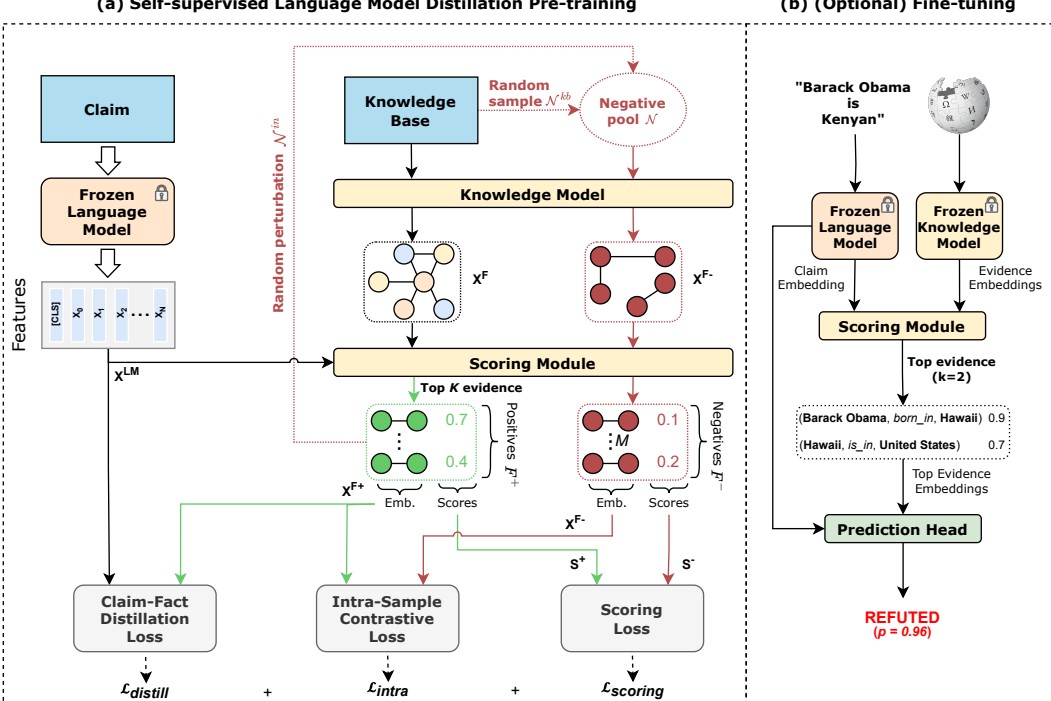

Figure 1: (a) A high-level overview of the SFAVEL framework. Given a textual claim, we use a frozen language model (orange box) to obtain its embedding features, $X^{LM}$. The knowledge base is fed to the knowledge model to produce a knowledge base embedding $X^F$. Then, the scoring module produces scores for facts in the knowledge base, conditioned upon the claim embedding. The positive sub-graph formed by the top $K$ facts is kept, denoted as $X^{F^+}$. Next, a negative pool of instances $\mathcal{N}$. Finally, both the positive and negative sub-graphs are encoded with the knowledge model, obtaining the positive and negative sub-graph embeddings, $X^{F^+}$ and $X^{F^-}$, and their respective scores, $S^+$ and $S^-$. Grey boxes represent three the different components of our self-supervised loss function used to train the knowledge model. (b) Optional supervised fine-tuning stage on a downstream task using the pre-trained model.

avoid the network finding trivial solutions where both positive and negative facts are given similar scores, a contrastive loss is used to encourage the separation, within the feature space, of features representing the positive and negative facts.

First, the knowledge model is initialized randomly and the backbone is initialized from any off-the-shelf pre-trained language model (e.g. a T5; Raffel et al. (2020a)). In this work, the backbone $f_L$ is kept frozen during the entire training and is only used to obtain a feature tensor for each claim, denoted as $X^{LM}$, which is used for distillation on the much smaller knowledge model. In order to obtain a single tensor representation per claim, we take the global average pooling (GAP) of the backbone features for each claim. For the knowledge model, we utilize a Relational Graph Attention Network (Busbridge et al., 2019).

Next, the knowledge model, $f_G : G \rightarrow \mathbb{R}^{|V| \times d_v}$, maps the input knowledge graph, $G$, into $X^{KB} \in \mathbb{R}^{|V| \times d_v}$, where $|V|$ is the number of nodes in $G$ and $d_V$ is the feature space dimensionality. In order to obtain a single feature tensor for each fact in $F$, we use a multilayer perceptron (MLP) that combines the head and tail embeddings for each fact into a single tensor, denoted as $X^F \in \mathbb{R}^{|F| \times d_F}$.

Then, given a single claim and fact embedding, denoted as $X_i^{LM}$ and $X_i^F$, respectively, we propose a score function, $f_{score}$. The goal of $f_{score}$ is to measure how likely it is that a given fact is in the same context as the corresponding claim. Specifically, the calculation of $f_{score}$ is defined as:

$$f_{\text{score}}(X_i^{LM}, X_i^F) = d(X_i^{LM}, X_i^F) \tag{1}$$

where $d(.)$ is a similarity score, which we take to be the $\mathbb{L}_2$ norm. Therefore, given the embedding of claim $x_i$, and every fact embedding in the knowledge base, $X^F$, we can compute the relevance scores $S_i = \{f_{\text{score}}(X_i^{LM}, X_j^F) \mid \forall \, j \in \{1, \ldots, |\text{F}|\}$. Then, the set of most relevant facts corresponding to claim $x_i$ is defined as:

$$F_i^+ = \text{top-rank}(S_i, K) \tag{2}$$

where top-rank$(\cdot, K)$ returns the indices of top $K$ items in a set. We can now obtain the embeddings of the positive facts, $X^{F^+} \subset X^F$, and the corresponding scores, $S^+ \subset S$, by using the indices of the top $K$ facts according to the scores $S$.

## 3.3 Generation of negative instances

In Section 3.2 we have described the process of obtaining both the positive fact embeddings and scores, $X^{F^+}$ and $S^+$, respectively. In this section we explain how to harness the graph structure of the knowledge base to produce corresponding negative signals for contrastive learning.

In order to produce a negative set for claim $x_i$, herein denoted as $\mathcal{N}_i$, we take inspiration from recent advances in graph contrastive learning (e.g. Xia et al. (2022); Yang et al. (2022); Rony et al. (2022)), and propose to generate two sets of negative instances: in-batch negatives and in-knowledge-base negatives, denoted as $\mathcal{N}_i^{in}$ and $\mathcal{N}_i^{kb}$, respectively. Our approach aims to generate negative samples that are factually false while preserving the contextual meaning of the entities, so that meaningful negative samples are fetched.

**In-batch negatives** To generate in-batch negatives we perform a random perturbation of the entities in the set of positive facts $F_i^+$ for a given claim $x_i$. Formally, let us define the set of triples in the positive set of claim $i$ as $T_i = \{(h_i, r_i, t_i) \mid \forall \, i \in \{1, 2, \ldots, |F_i^+|\}$, where $h$, $r$, $t$ represents head, relation and tail, respectively. Our goal is to generate $\mathcal{M}$ negative samples in each given batch $\mathcal{B}$. For each triple $t_{h,r,t}$ in $T_i$, we decide in a probabilistic manner whether the perturbation is done on the head or the tail of the triple. For this, let us define a random variable $perturb\_head$ $\sim \text{Bern}(p_{head})$ sampled from a Bernoulli distribution with parameter $p_{head}$, dictating whether the head of a triple should be perturbed, with probability $p_{head}$, or the tail otherwise. Then for each triple $t_{h,r,t}$, we generate a negative triple $t_{h',r,t}$ or $t_{h,r,t'}$, by altering head ($p_{head} = 1$) or tail ($p_{head} = 0$), respectively, such that the new head, $h'$, or tail, $t'$, are sampled from $\varepsilon$ uniformly. To provide semantically meaningful negatives, we enforce the entity type of the randomly sampled head/tail to be of the same type as the one in $t_{h,r,t}$.

**In-knowledge-base negatives** Given the nature of the in-batch negative generation process, the negative triples are bounded to be semantically similar to the corresponding positive triples, and hence close by in the feature space. Therefore, this bias leads to under-exploration of other parts of the knowledge base feature space. In order to alleviate this issue, we propose to add randomly-sampled facts from the knowledge base into the negative set. Specifically, given the knowledge base $G$, we sample $\mathcal{M}$ triples that are at least $H$ hops away from $F_i^+$. This encourages the negative generation procedure to dynamically explore other parts of the knowledge base.

To obtain the final negative set for claim $x_i$, we join both the set of in-batch negatives $\mathcal{N}_i^{in}$ and in-knowledge base negatives $\mathcal{N}_i^{kb}$ as:

$$\mathcal{N}_i = \mathcal{N}_i^{in} \cup \mathcal{N}_i^{kb} \tag{3}$$

Finally, we obtain the embeddings of the negative set, $X^{F^-}$ from the knowledge model, and the negative scores from the scoring module as $S^- = \{f_{\text{score}}(X_i^{LM}, X_j^{F^-}) \mid \forall \, j \in \{1, 2, \ldots, |N_i|\}$.

## 3.4 Claim-Fact Matching via Language Model Distillation

Once we get positive and negative fact embeddings and scores, they can be used for the distillation process. In particular, we seek to learn a low-dimensional embedding that "distills" the feature

correspondences of a pre-trained language model between textual claims and the knowledge base features produced by the knowledge model. To achieve this, we propose a loss function composed of 3 terms: claim-fact distillation, $\mathcal{L}_{distill}$, intra-sample contrastive loss, $\mathcal{L}_{intra}$, and scoring loss, $\mathcal{L}_{scoring}$.

**Claim-Fact Distillation**    To transfer the feature correspondences from the pre-trained language model to the knowledge model, we propose a feature-based (Zagoruyko & Komodakis, 2017) claim-fact distillation loss. Specifically, we propose the following distillation loss:

$$\mathcal{L}_{distill} = \sum_{j \in F^+} \left\| \frac{F_j^{KM}}{\|F_j^{KM}\|_2} - \frac{F_j^{LM}}{\|F_j^{LM}\|_2} \right\|_p^p. \tag{4}$$

where $F_j^{KM}$ and $F_j^{LM}$ are respectively the knowledge model (student) and language model (teacher) feature representations, for each fact $j$, in the positive fact set, $F^+$. $p$ refers to the norm type, and we use $p = 2$ for $\mathbb{L}_2$-normalized features. To obtain $F_j^{LM}$, we compute the pooled features representation of the verbalized triplet $j$ given by the LM, similarly to Lu et al. (2022). Therefore, the distillation is computed as the Mean Squared Error on the normalized fact embeddings.

**Intra-Sample Contrastive Loss**    The intra-sample distillation loss derives from the standard contrastive loss. The aim of the contrastive loss is to learn representations by discriminating the positive instance, among negative samples. For instance, in MoCo (He et al., 2020), two views, $x$ and $x^{'}$, of one input image are obtained using augmentation, and an encoder $f_q$ and momentum encoder $f_k$ are used to generate embeddings of the positive pairs, such that $q = f_q(x)$ and $k = f_k(x^{'})$. In this case, the contrastive loss can be defined as:

$$\mathcal{L}_{contrastive} = -\log \frac{\exp(\mathbf{q} \cdot \mathbf{k}^+/\tau)}{\sum_{i \in N} \exp(\mathbf{q} \cdot \mathbf{k_i}/\tau)}. \tag{5}$$

Similarly to Frosst et al. (2019), we extend the contrastive loss function by replacing the query, $q$, in the original formulation with the claim embedding, denoted as $X_i^{LM}$, and extending the formulation to account for multiple positive keys. Consequently, we contrast the query with respect to each of the individual positive (numerator) and negative (denominator) facts, as follows:

$$\mathcal{L}_{intra} = \frac{-1}{|F^+|} \sum_{i \in F^+} \log \frac{\exp(X_i^{LM} \cdot X_i^{F^+}/\tau)}{\sum_{j \in F^-} \exp(X_i^{LM} \cdot X_j^{F^-}/\tau)}. \tag{6}$$

where $\tau$ is the temperature parameter, used during training to smooth the logits distribution. The rationale of $\mathcal{L}_{intra}$ is to pull the positive fact embeddings close to the claim representation while simultaneously pushing away the negative fact embeddings.

**Scoring Loss**    The scoring loss is a variant of the conventional pair-wise ranking loss (Chen et al., 2009). Ranking losses are used to evaluate the performance of a learned ranking function. In this work, we propose the $\mathcal{L}_{scoring}$ loss function to maximize the scores given by the scoring model to the positive facts, $F^+$, and minimize the scores of the negative facts, $F^-$, for a given claim $x_i$. In particular, we minimize the following loss:

$$\mathcal{L}_{scoring} = \sum_{i \in F^+} \sum_{j \in F^-} max\left(0, \gamma + f_{\text{score}}(X_i^{LM}, X_i^{F^+}) - f_{\text{score}}(X_i^{LM}, X_j^{F^-})\right) \tag{7}$$

where $\gamma$ is a margin factor. Minimizing $\mathcal{L}_{scoring}$ encourages elements of $f_{\text{score}}(X_i^{LM}, X_i^{F^+})$ to be highly ranked and elements of $f_{\text{score}}(X_i^{LM}, X_j^{F^-})$ to have low scores. More explicitly, it optimizes the model parameters so that the scores of positive facts, $(h, r, t) \in F^+$, are higher than the scores of negative facts $(h', r', t') \in F^-$.

Finally, SFAVEL's full loss is:

$$\mathcal{L}_{total} = \lambda_{distill}\mathcal{L}_{distill} + \lambda_{intra}\mathcal{L}_{intra} + \lambda_{scoring}\mathcal{L}_{scoring} \qquad (8)$$

where $\lambda_{distill}$, $\lambda_{intra}$, $\lambda_{scoring} \in \mathbb{R}$. In practice, we found that a ratio of $\lambda_{intra} \approx \lambda_{scoring} \approx 2\lambda_{distill}$ led to good experimental results.

## 4 EXPERIMENTS

In this section, we present a comparative study of the results of our proposed method on standard benchmarks for fact verification, as well as ablation studies on the most relevant components. We first describe the datasets, evaluation and training settings. Next, we discuss extensive experiments on our method for the task of fact verification. Finally, we run a set of ablation studies to evaluate the impact of the most important components of our proposed framework.

### 4.1 IMPLEMENTATION DETAILS

**Datasets and evaluation** We use the FEVER (Thorne et al., 2018) dataset for all our experiments and comparison against previous methods. For pre-training we use the official FEVER training set. For providing the performance comparisons against previous work, we use the official FEVER testing set. In our ablation studies, we employ the official FEVER development split. To evaluate the learning performance in a low-data regime, we randomly sample 1%, 5% or 10% of the training data. As knowledge base, we use the Wikidata5m (Wang et al., 2021b). We provide some examples of claims from FEVER in Section A.1 of the Appendix. Finally, we also compare on the FB15k-237 dataset from (Toutanova et al., 2015) in Section A.2 of the Appendix.

**Pretraining** Eight models with a variety of sizes are used as pre-trained language models: T5-Small (Raffel et al., 2020a), DeBERTaV3 (He et al., 2023b), XLNet (Yang et al., 2020), RoBERTa (Liu et al., 2019), BERT (Devlin et al., 2019), Transformer-XL (Dai et al., 2019) and GPT-2 (Radford et al., 2019). The pre-trained language models are kept frozen during pre-training with our method. The officially released weights from HuggingFace (Wolf et al., 2020) are used to initialize the pre-trained language models for fair comparisons. Pre-training is run for a total of 1000 epochs. During training we use a RGAT as the knowledge model with 3 convolutional layers, with a hidden size of 512. The projector from node embeddings to triple embeddings is a MLP with the same dimensionality as the pre-trained language model sentence embedding size. The model is trained with the SGD optimizer with momentum 0.9 and weight decay 0.0001. The batch size is set to 512 over 4 A100 GPUs, and the coefficients for the different losses are $\lambda_{intra} = \lambda_{scoring} = 1$, $\lambda_{distill} = 2$. We set the temperature $\tau$ = 0.1. We use $K = 5$ for the number of facts to keep after scoring. The number of negative instances used in the negative pool for contrastive learning is set to $M = 4096$.

**Linear Probe** In order to evaluate the quality of the distilled claim-fact matching features, we follow common evaluation protocols (Gansbeke et al., 2021; Chen et al., 2020d) for measuring transfer learning effectiveness. Specifically, we train a linear classifier to perform label assignment to claims (see Figure 1b for an example illustration). The classifier is trained for 200 epochs, using the SGD optimizer with 20 as the initial learning rate. The only purpose of this linear probe is to evaluate the quality of the features and is not part of the SFAVEL pretraining procedure.

### 4.2 RESULTS

We summarize our main results on the FEVER fact verification benchmark in Table 1. Our method significantly outperforms the prior state of the art, both supervised and unsupervised. In particular, SFAVEL improves by +9.23% on label accuracy in the test set when using a simple linear probe and a frozen backbone pre-trained using our method. Notably, even though our method has been trained without any data annotations, it is capable of outperforming the best supervised method (ProoFVer) by +9.58% label accuracy. These experiments demonstrate the benefits of our task-specific unsupervised pretraining framework for learning rich feature representations for claim-fact matching.

Table 1: Performance on the FEVER benchmark (label accuracy and FEVER score in %) of our proposed pre-training approach after fine-tuning a linear classification probe on the FEVER benchmark. In this experiment we use the GPT-2-XL as backbone. Underlined performances indicate the top score for a particular metric, bold indicates the overall best method. We show that our SFAVEL outperforms previous methods, both supervised and unsupervised. Dataset splits are from (Thorne et al., 2018).

| Method | Unsupervised | Dev | | Test | |
|---|---|---|---|---|---|
| | | LA | Fever Score | LA | Fever Score |
| GEAR (Zhou et al., 2019) | ✗ | 74.84 | 70.69 | 71.60 | 67.10 |
| KGAT (Liu et al., 2021) | ✗ | 78.29 | 76.11 | 74.07 | 70.38 |
| Di Liello et al. (2022) | ✓ | 81.21 | - | 74.39 | - |
| GERE (Chen et al., 2022b) | ✗ | 79.44 | 77.38 | 75.24 | 71.17 |
| CorefBERT (Ye et al., 2020) | ✗ | - | - | 75.96 | 72.30 |
| DREAM (Zhong et al., 2020) | ✗ | 79.16 | - | 76.85 | 70.60 |
| ProoFVer (Krishna et al., 2021) | ✗ | 80.74 | 79.07 | 79.47 | 76.82 |
| Jobanputra (2019) | ✓ | 80.20 | - | 80.25 | - |
| SFAVEL (Ours) | ✓ | **90.32** | **88.10** | **89.48** | **86.15** |

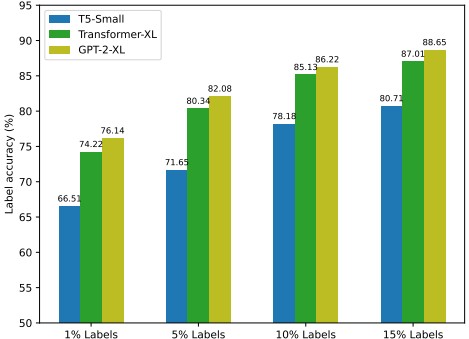

Figure 2: Low-data experiments by fine-tuning on 1%, 5%, 10% and 15% of data over 3 different language backbones.

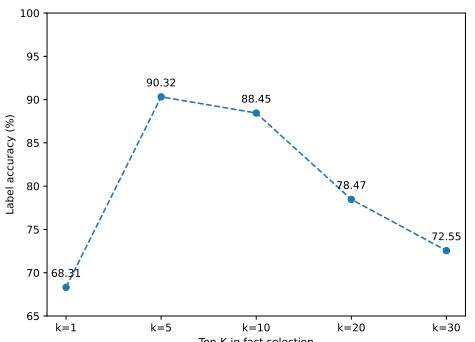

Figure 3: Label accuracy with different $K$ for number of facts selected after scoring.

Furthermore, following previous works in contrastive learning (Chen et al., 2020b), we evaluate the proposed method by distilling 3 different language model backbones (T5-Small, Transformer-XL, GPT-2-XL) and fine-tuning in a low-data setting by using 1%, 5%, 10% and 15% of labeled data. As shown in Figure 2, our method is capable of achieve on-par performance with recent methods despite only fine-tuning with 1% of the data, reaching 74.22% and 76.14% dev set accuracy with Transformer-XL and GPT-2-XL backbones, respectively. When using 5% of labelled data, SFAVEL surpasses previous state-of-the-art on the FEVER benchmark. This experiment highlights the high-quality features our framework is capable of learning for claim-fact matching, allowing high accuracy even when only a few labelled data points are available.

## 4.3 ABLATION STUDIES

In the following section, we provide several ablation studies for our proposed approach. All experiments and results are performed on the FEVER development set with the GPT-2-XL as language model backbone unless explicitly stated.

**Pre-trained Language Model** To understand the impact of pre-trained language model selection for distillation backbone, we perform an ablation study and report the results in Table 2. We analyze the effect of using several different language models in SFAVEL, such as T5-Small, DeBERTaV3, XLNet, GPT-2-Small, RoBERTa, BERT, Transformer-XL and GPT-2-XL. We choose this particular set of language models as they are diverse in terms of their number of parameters. The smallest language model in our experiments is T5-Small (60 Million parameters), with the biggest LM being GPT-2-XL (1.5 Billion parameters). This gives some insight into how the language representation capabilities of each of the models affects the distillation effectiveness when using SFAVEL. We find

Table 2: Accuracy of linear classification results on FEVER dev set using different pretrained language models as backbone for distillation.

| Backbone | # Params | Accuracy |
|---|---|---|
| T5-Small | 60M | 80.79 |
| DeBERTaV3 | 86M | 82.09 |
| XLNet | 110M | 84.46 |
| GPT-2-Small | 117M | 84.92 |
| RoBERTa | 125M | 85.31 |
| BERT | 140M | 87.92 |
| Transformer-XL | 257M | 89.51 |
| GPT-2-XL | 1.5B | **90.32** |

Table 3: Effect of the different components of our loss function. From left to right: language model backbone used, whether claim-fact distillation, intra-sample contrastive and scoring losses are used, respectively.

| Backbone | Distill Loss | Cont Loss | Scoring Loss | Accuracy |
|---|---|---|---|---|
| T5-Small | | ✓ | ✓ | 56.34 |
| T5-Small | ✓ | | ✓ | 72.71 |
| T5-Small | ✓ | ✓ | | 77.46 |
| T5-Small | ✓ | ✓ | ✓ | 80.79 |
| GPT-2-XL | | ✓ | ✓ | 57.20 |
| GPT-2-XL | ✓ | | ✓ | 79.64 |
| GPT-2-XL | ✓ | ✓ | | 87.16 |
| GPT-2-XL | ✓ | ✓ | ✓ | **90.32** |

that the GPT-2-XL is the best feature extractor of the list and leads by a significant margin in terms of accuracy. However, we note that even the smallest backbone (T5-Small; 60M parameters), although modestly, achieves performance greater than the previous state-of-the-art (+0.54% accuracy).

**Influence of $K$ in fact selection**  We inspect the impact of $K$ in fact selection after scoring for model performance. As shown in Figure 3, the results are consistent for a range of $K$ ($K = 1, 5, 10, 20, 30$). In particular, we observe a decrease in classification accuracy with $K = 10, 20, 30$ compared with $K = 5$. We believe this decrease is caused by the factual noise introduced when $K$ becomes large, where irrelevant information is used for verifying the specific claim. In contrast, with $K = 1$, the performance drop is caused by a lack of information, as only a single fact is used to check a claim. Avoiding this is critical in settings where multiple pieces of evidence are required for reasoning, as it is for FEVER.

**Loss function components**  We evaluate the different loss functions described in Section 3.4, and provide the results in Table 3. In particular, we investigate suppressing particular components of the loss function, such as the claim-fact distillation loss, intra-sample contrastive loss, and scoring loss. To do so, we set their respectively $\lambda$ factors to 0, effectively nullifying their influence during training. We find that these loss components lead to significant performance decreases when removed, therefore justifying our architectural decisions.

## 5 CONCLUSION

This paper proposes a new self-supervised pretraining method based on distillation, named SFAVEL, which aims to produce high-quality features for claim-fact matching in the context of fact verification tasks. We have found that modern self-supervised language model backbones can be distilled into smaller knowledge-aware models to yield state-of-the-art fact verification performance. Our approach achieves this by introducing a novel contrastive loss, that leverages inductive biases in the fact verification task, and exploits them for accurate and entirely unsupervised pretraining for claim-fact matching. We show that SFAVEL yields a significant improvement over prior state-of-the-art, both over unsupervised and supervised methods, on the FEVER fact verification challenge (+8% accuracy) and the FB15k-237 dataset (+5.3% Hits@1). Finally, we justify the design decisions of SFAVEL by performing ablation studies over the most important architectural components. The proposed self-supervised framework is a general strategy for improving unsupervised pretraining for fact verification, and we hope it will guide new directions in the unsupervised learning field.

REPRODUCIBILITY STATEMENT

To guarantee reproducibility of this paper, we release the source code at https://github.com/AdrianBZG/SFAVEL.

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

# A APPENDIX

## A.1 EXAMPLES OF CLAIMS IN THE FEVER DATASET

For illustration purposes, in Table 4 we provide some examples of claims, with the annotated evidence and labels, from the FEVER dataset.

| Claim | Evidence | Label |
|---|---|---|
| The Rodney King riots took place in the most populous county in the USA | (1992 Los Angeles riots, also_known_as, Rodney King riots), (Rodney King riots, occurred_in, Los Angeles County), (Los Angeles County, county_of, USA), (Los Angeles County, most_populous_county_of, USA) | SUPPORTED |
| Giada at Home was only available on DVD | (Giada at Home, is_a, Televion Show), (Giada at Home, aired_on, Food Network), (Food Network, is_a, Television Channel) | REFUTED |
| Al Jardine is an American rhythm guitarist | (Alan Charles Jardine, is_a, guitarist), (Al Jardine, alias_of, Alan Charles Jardine), (Alan Charles Jardine, has_nationality, American) | SUPPORTED |
| Bob Ross created ABC drama The Joy of Painting | (Robert Norman Ross, also_known_as, Bob Ross), (Robert Norman Ross, is_a, Painter), (Robert Norman Ross, is_a, Television host), (Robert Norman Ross, creator_of, The Joy of Painting), (The Joy of Painting, is_a, Instructional television program) | REFUTED |

Table 4: Examples from FEVER (Thorne et al., 2018) of claims, their annotated evidence and label.

## A.2 ADDITIONAL EXPERIMENTS ON THE FB15K-237 DATASET

In addition to our evaluations in Section 4.2 we compare SFAVEL to prior art on the FB15k-237 fact checking task presented in (Toutanova et al., 2015). We utilize the same evaluation metrics and dataset processing as (He et al., 2023a). In this setting, in contrast to FEVER evaluation, we obtain the top $K$ facts given a claim directly from SFAVEL's unsupervised pre-trained model, without the need for a downstream prediction head. In Table 5 we find that SFAVEL is able to achieve +5.3%, +3.6% and +6.3% for Hits@1, Hits@3 and Hits@10, respectively, compared to the previous state of the art.

| Method | Hits@1 | Hits@3 | Hits@10 |
|---|---|---|---|
| KG-BERT (Yao et al., 2019) | - | - | 42.0 |
| StAR (Wang et al., 2021a) | 20.5 | 32.2 | 48.2 |
| KG-S2S (Chen et al., 2022a) | 25.7 | 37.3 | 49.8 |
| SimKGC (Wang et al., 2022) | 24.6 | 36.2 | 51.0 |
| MoCoSA (He et al., 2023a) | 29.2 | 42.0 | 57.8 |
| SFAVEL (Ours) | **34.5** | **45.6** | **64.1** |

Table 5: Performance on the FB15k-237 dataset test split. We use the GPT-2-XL as backbone for distillation. Following previous methods, we measure performance using the Hits@1, 3, 10 metric. We obtain the top $K$ facts from our unsupervised pre-trained model, without the need for a downstream prediction head.

