# OpenReview forum: "Unsupervised Pretraining for Fact Verification by Language Model Distillation"
_ICLR.cc/2024/Conference — ICLR 2024 poster_

### Official Review · Reviewer_kcpN · 2023-10-31

**Soundness:** 3 good
**Presentation:** 2 fair
**Contribution:** 2 fair
**Rating:** 6
**Confidence:** 4

**Summary:**

This work proposes an unsupervised framework for unsupervised fact verification. Specifically, three losses are designed to encourage the two models adopted in this framework to produce high-quality features for claim-fact matching. The authors conduct experiments on the FEVER dataet. Experimental results show that the proposed method can achieve good performance on the FEVER dataset.

**Strengths:**

1. Experimental results on the FEVER dataset show the advantage of the proposed method. The improvement seems very significant.

2. The unsupervised manner of conducting fact verification is encouraged and useful.

**Weaknesses:**

1. Though the experiments show the effectiveness of the method, I do not get how the framework solves the cold starting. For the scoring module, the claim embeddings from the LM are very different from those of Knowledge from the knowledge model. Then how does the framework pick the top-k evidence at the beginning? How does $L_{distill}$ work at the early iterations? This is an important prerequisite that should be clearly stated in the paper.

2. Why does the paper only evaluate the effect of each loss on a smaller T5 model? Considering the best performance reported in the work is based on Transformer-XL, ablation studies based on it are desired.

3. The annotation in the methodology section makes me really confused. For example but not limited to:

    (1) In Section 3.1, what does the V and each $v_i$ mean? I cannot get it until I read through the whole methodology section.

    (2) The use of subscript and superscript is messed up. I think embedding is presented as $X_F$ in Section 3.2, but it becomes $X^F$ in Section 3.3.

    (3) Some annotations are not really necessary, e.g., $N_i$ in Equation 3 are not used anywhere else in the paper.

    I encourage the authors to make Section 3 more concise and clear by better formula presenting.

**Questions:**

Please refer to the questions above in the weakness. Especially please make it clear how the model works at the beginning when the representations of the LM and the knowledge model are significantly different.

---

> ### Author Response · Authors · 2023-11-17
> **Responding to Reviewer kcpN**
>
> We thank you for your time and for providing constructive feedback. We have provided a high-level overview of major concerns in the general rebuttal and address more targeted questions in this comment.
>
> ## Additional Experiments on Loss Function Ablation Study
>
> We have updated Table 3 to include the loss function ablation experiments on the best performing backbone (GPT-2-XL) as well as a different, smaller backbone (T5-Small). This allows for a comprehensive understanding of the most critical components of the loss function across two different backbone settings. We consistently observed that the distillation loss is the most relevant component of the loss function, and the contrastive and scoring loss components allow for further improvement in the quality of the learned embeddings.
>
> ## Clarification on Distillation Loss and Cold Start
>
> We appreciate your attention to the issue of cold start and your request for clarification on how our framework addresses it. Our method indeed takes the cold start problem into consideration, where during the early training stages, the knowledge model provides completely dissimilar embeddings to triplets in comparison to the claim embedding given by the language model. As you correctly pointed out, we tackle this with the $\mathcal{L}\_{distill}$ component in the loss function. We would like to clarify how this works by providing a fine-grained explanation of $\mathcal{L}\_{distill}$.
>
> The $\mathcal{L}\_{distill}$ is defined in Section 3.4 as follows:
>
> $
> \begin{equation}
> {\mathcal{L}\_{distill}} = \sum\_{j\in F^{+}} || \frac{F^{KM}\_j}{||F^{KM}\_j||_2} - \frac{F^{LM}\_j}{||F^{LM}\_j||_2} ||_p.
> \end{equation}
> $
>
> In the above equation, we have two main elements: $F^{KM}\_j$ and $F^{LM}\_j$, corresponding to the embeddings obtained from the knowledge model and the language model, respectively, for each fact, $j$, in the positive set $F^{+}$. To obtain $F^{LM}_j$, we generate embeddings for the verbalised fact triplets in the knowledge graph using a language model (e.g. GPT-2-XL). These embeddings capture the fact's textual representation and provide a way to compare it to the structured knowledge from the knowledge graph. This way of obtaining triplets embeddings from sequence-based models is standard in the state-of-the-art (see [1] for a recent example). These embeddings are computed only once during data preprocessing and stored to be used during training.
>
> Therefore, during training, the $\mathcal{L}\_{distill}$ encourages the knowledge model to embed the triplets in the feature space as close as possible to the corresponding verbalised triplet embedding from the language model. As shown in the ablation study in Table 3, this loss component is critical for the performance of our method, as without it the model can not solve the cold start. Moreover, this is consistent with our findings at the end of Section 3.4, where we state that weighting more $\mathcal{L}\_{distill}$ than $\mathcal{L}\_{contrastive}$ or $\mathcal{L}\_{intra}$ is the best setting for optimal pre-training results.
>
> To make this more clear, we have extended our explanation in the Claim-Fact Distillation Loss paragraph in Section 3.4.
>
> ## Notation Typos and Simplification of Section 3
>
> We appreciate you pointing out these typos and have made another pass through the work to correct errors. In particular, we have made the notation in Section 3 consistent throughout the paper, fixed notation typos and made it more clear and concise when possible.
>
> Thank you again for helping us improve this work.
>
> Please let us know if you would like to see additional experiments or have further feedback.
>
> ## References
>
> [1] Lu et al., “KELM: Knowledge Enhanced Pre-Trained Language Representations with Message Passing on Hierarchical Relational Graphs”. ICLR, 2022.

---

> > ### Author Response · Authors · 2023-11-20
> > **A gentle reminder**
> >
> > Dear Reviewer kcpN,
> >
> > We sincerely appreciate your time and effort in reviewing our paper. We believe that your comments would further strengthen our paper. During the discussion period, we have made every effort to faithfully address all your comments in the responses and revision. Therefore, we we would appreciate if you could read our responses and reconsider your rating on our work.
> >
> > We thank you again for your time and efforts in reviewing our paper.
> >
> > Please let us know if you have any further questions.
> >
> > Best regards,
> > Authors.

---

### Official Review · Reviewer_GMHM · 2023-10-31

**Soundness:** 3 good
**Presentation:** 2 fair
**Contribution:** 3 good
**Rating:** 6
**Confidence:** 3

**Summary:**

The paper proposes an unsupervised method called SFAVEL for fact verification. The method aim is to distil knowledge from a language model into a knowledge model so that they produce similar vectors for fact in a formal form and in a natural language form. Besides to make the knowledge model distinguish facts, contrastive learning is employed. The paper shows that SFAVEL remarkably outperforms SoTA fact-verification models on the FEVER dataset (about 8% label accuracy higher than the best model in the literature).

**Strengths:**

The experimental result of SFEVEL on the FEVER dataset is remarkable.

**Weaknesses:**

First of all, the paper is confusing in using the term "unsupervised". The proposed method SFAVEL is unsupervised because it is for learning a knowledge model. However, the fact-verification model reported in section 4 is supervised. The model uses SFAVEL for mapping fact / claim to vectors and then uses a classifier trained in a supervised learning manner.

Secondly, it is unclear about what is the used "linear probe". Fig 4b shows that the linear probe takes top evidence as input. But then how can we verify the input claim if we use only evidence (and their scores)? E.g. how knowing "Obama was born in Hawaii" and "Hawaii is in the US"  can reject a claim without knowing what the claim is?

Thirdly, although the performance of the proposed model is remarkable, it is unclear why there's such a big gap between it and the existing models in the literature. What are cases that the proposed model can solve but the others can? Does the model find some crucial factors that the others miss?

Last but not least, the proposed SFAVEL is for learning a knowledge model. But it is unclear whether that knowledge model is useful for other fact-verification cases (like on other FEVER dataset -- FEVER 2.0 for example). Also, whether that knowledge model is also useful for other downstream task requiring fact?

**Questions:**

In contrastive learning loss, e.g. eq 5, the denominator includes the numerator. However, it's not the case in eq 6. Why is that? What is the impact of excluding the numerator out of the denominator?

In section 4.2, what would happen if more data are used (rather than only 1, 5, 10%)?

In table 3, as transformer-XL is used in the end, why the ablation is for T5-small instead?


-------
The score is updated after reading the authors' response.

---

> ### Author Response · Authors · 2023-11-17
> **Responding to Reviewer GMHM (Part 1/2)**
>
> We thank the reviewer for the helpful comments, and would like to clarify the concerns raised. We have provided a high-level overview of major concerns in the general rebuttal and address more targeted questions in this comment.
>
> # Response to Weaknesses:
>
> ## Experiments on Additional Datasets
>
> As this is a common question between some reviewers, we have provided experiments on additional datasets in the general rebuttal. In particular, we benchmark SFAVEL's unsupervised pre-trained model in the FB15k-237 dataset, which is another common benchmark for fact checking tasks. When compared against state-of-the-art methods, our approach is able to achieve +5.3%, +3.6% and +6.3% for Hits@1, Hits@3 and Hits@10 for fact ranking, respectively.
>
> ## On Performance Gap Between Previous Methods and SFAVEL
>
> In our work, we leverage self-supervised learning to exclusively learn and understand the underlying structure of the claim-fact alignments, leading to high-quality embeddings that can be used in further fact-related downstream tasks, as shown in our new experiments. This is different to the approach taken by previous works (e.g. [1, 2, 3], to mention some), which aim to learn models for both feature representation and prediction at the same time, in a supervised setting. This leads to models that focus on learning specific and potentially spurious patterns associated with the labels, without encouraging the model to learn rich, meaningful and general representations of the data. For this reason, SFAVEL is able to learn intrinsic properties of the data that attain better transferability.
>
> ## Clarification on Terminology: Proposed Unsupervised Pre-training Method vs Supervised Linear Probe
>
> We would like to emphasise that the main contribution of our work is the unsupervised (self-supervised) pre-training framework. The design of our method is motivated by the goal of learning high-quality claim-fact embeddings useful for downstream fact verification tasks. The embeddings obtained with SFAVEL can be used as input for training any downstream model on different tasks related to fact checking. This is a standard practice in self-supervised learning (see e.g. [4, 5]).
>
> In our experiments we use a linear probe, which is just a linear projection layer, as an example of a downstream model using SFAVEL’s claim embedding and facts pooled embedding as input for claim classification. This is required for the case of FEVER, as the task is to classify claims, but it is not for FB15k-237, where only ranking of claims given a fact is important. On the latter, we do not need a supervised stage, and only use the ranking given by SFAVEL directly for evaluation. The aim of this experiment is two-fold: first, we show the generalizability of our unsupervised pre-training method on fact verification tasks with different nature, and second we obtain state-of-the-art performance with a simple linear layer classifier relying only on pre-trained SFAVEL features, showing the quality of the learned embeddings.
>
> We have now clarified this with the following changes: we added “Pretraining” in the paper title, clarified the optionality of the downstream task in Figure 1, and added more details in the explanation for the Linear Probe description in Section 4.1.

---

> > ### Author Response · Authors · 2023-11-17
> > **Responding to Reviewer GMHM (Part 2/2)**
> >
> > # Response to Questions:
> >
> > ## Clarification on Intra-Sample Contrastive Loss Formulation
> >
> > To clarify the intra-sample contrastive loss proposed in our paper, we provide below a step-by-step explanation on how to get from the InfoNCE contrastive loss to the SFAVEL intra-sample contrastive loss.
> >
> > The original formulation of the InfoNCE [6] contrastive loss is as follows:
> >
> > $
> > \begin{equation}
> > {\mathcal{L}\_{contrastive}} =  - \log \frac{{\exp (\mathbf{q} \cdot {\mathbf{k^+} }/\tau )}}{{\sum\_{i\in N} \exp (\mathbf{q} \cdot {\mathbf{k\_i} }/\tau )}}
> > \end{equation}
> > $
> >
> > In the above formula, the main elements are the query embedding ($q$), one positive key embedding ($k^+$), and the embeddings for multiple negative keys ($k_i$, for every negative instance, $i$, in the set $N$). The only common element between the numerator and the denominator is the query ($q$). The objective is to pull the query and positive key embeddings together, whilst pushing away the query embedding with respect to the negative keys.
> >
> > In our setting, we propose several changes. First, in both numerator and denominator, we replace the query ($q$) by the mean pooling of the positive facts (denoted as $\hat{X}^{F^{+}}$). Second, in the denominator, the negative keys (each $k_i$ in the formula above) are replaced by each of the negative fact embeddings, as follows:
> >
> > $\sum\_{j \in F^{-}} \exp (\hat{X}^{F^{+}} \cdot {X\_{j}^{F^{-}}}/\tau )$
> >
> > Finally, since we have multiple positive facts in a batch, and the original InfoNCE loss only takes one positive key into account, we adapt the numerator to introduce multiple positive keys, similar to [7]. Therefore, the numerator becomes the sum of similarities between the query and the positive facts, scaled by $\tau$, as follows.
> >
> > $\sum\_{i \in F^{+}} \exp (\hat{X}^{F^{+}} \cdot {X\_{i}^{F^{+}}}/\tau )$
> >
> > When put together, it becomes our intra-sample contrastive loss.
> >
> > ## Clarification and Additional Experiments on Low-Data Settings
> >
> > In our low-data setting, we follow the classic supervised fine-tuning procedure for evaluating a pretraining method in terms of feature quality (see e.g. [1] for a recent work also doing this). The rationale is to understand how good the pre-training is by using the amount of data required for supervised fine-tuning as a proxy. Typically a ratio of 1%, 5% and 10% is used, as we do in our paper. As per your request, we have now since updated the experiments by using the GPT-2-XL backbone, and also evaluated the case of 15% of data for training. We observe that the accuracy obtained during fine-tuning with 15% of the data is 88.65%. As this accuracy is close to the full-data (80% training split) case (90.32%), we can say that the feature representations learned by SFAVEL during the pretraining stage are of enough quality to lead to a significant reduction in terms of required data for supervised training on the FEVER dev dataset.
> >
> > ## Additional Experiments on Loss Function Ablation Study
> >
> > We have updated Table 3 to include the loss function ablation experiments on the best performing backbone (GPT-2-XL) as well as a different, smaller backbone (T5-Small). This allows for a comprehensive understanding of the most critical components of the loss function across two different backbone settings. We consistently observed that the distillation loss is the most relevant component of the loss function, and the contrastive and scoring loss components allow for further improvement in the quality of the learned embeddings.
> >
> > Please let us know if you would like to see additional experiments and thank you again for helping us improve this work.
> >
> > ## References
> >
> > [1] Krishna et al., "ProoFVer: Natural Logic Theorem Proving for Fact Verification". TACL, 2021.
> >
> > [2] Zhong et al., “Reasoning Over Semantic-Level Graph for Fact Checking”. ACL, 2020.
> >
> > [3] Chen et al., “GERE: Generative Evidence Retrieval for Fact Verification”. SIGIR, 2022.
> >
> > [4] Bouniot et al., “Proposal-Contrastive Pretraining for Object Detection from Fewer Data”. ICLR, 2023.
> >
> > [5] Hamilton et al., “Unsupervised Semantic Segmentation by Distilling Feature Correspondences”. ICLR, 2022.
> >
> > [6] van den Oord et al., “Representation Learning with Contrastive Predictive Coding”. arXiv:1807.03748, 2019.
> >
> > [7] Frosst et al., “Analyzing and Improving Representations with the Soft Nearest Neighbor Loss”. ICML, 2019.

---

> > > ### Author Response · Authors · 2023-11-20
> > > **A gentle reminder**
> > >
> > > Dear Reviewer GMHM,
> > >
> > > We sincerely appreciate your time and effort in reviewing our paper. We believe that your comments would further strengthen our paper. During the discussion period, we have made every effort to faithfully address all your comments in the responses and revision. Therefore, we we would appreciate if you could read our responses and reconsider your rating on our work.
> > >
> > > We thank you again for your time and efforts in reviewing our paper.
> > >
> > > Please let us know if you have any further questions.
> > >
> > > Best regards,
> > > Authors.

---

> > > > ### Comment · Reviewer_GMHM · 2023-11-22
> > > > **Reply**
> > > >
> > > > I would like to thank the authors for the thoughtful response, which addresses most of the weaknesses raised in my review. I thus raise the score to 6.
> > > >
> > > > However, because of the newly added experimental result in A2, I think that the main aim of the paper becomes unclear. More specifically, the experiment in A2 is for knowledge base completion rather than fact verification. Therefore, if the aim is still fact verification, I would recommend an experiment on fact-verification datasets. Otherwise, the paper can aim at knowledge base model pretraining and should present experiments on related experiments.
> > > >
> > > > Re infoNCE, the authors can check in the original paper, equation 4, that the numerator includes N examples, one of them is positive. However, the numerator used in the paper doesn't include the positive example. My question in the review is about that difference.

---

> > > > > ### Author Response · Authors · 2023-11-23
> > > > > **Thank you**
> > > > >
> > > > > Dear Reviewer GMHM,
> > > > >
> > > > > We really appreciate the time you spent reviewing our work and going through our rebuttal, and are very thankful for your support of our work.
> > > > >
> > > > > Regarding the main aim of the paper and experimental results in A.2, we belive this does not deviate from the way other works on fact verification / fact checking have been traditionally evaluated, and we point to a few example papers in our rebuttal. For this reason, we believe the main aim of the paper is clear: a new unsupervised pretraining method applied to fact verification downstream tasks, via language model distillation.
> > > > >
> > > > > Re. InfoNCE and our formulation of the claim-fact distillation loss, we would like to point out that our numerator includes the positive query, denoted as $\hat{X}^{F^{+}}$, as well as in the denominator. In the numerator the query is contrasted with the positive keys, and in the denominator it is contrasted with the negative keys.
> > > > >
> > > > > We thank you again for your time and efforts in reviewing our paper, which has significantly improved its content.
> > > > >
> > > > > Please let us know if you have any further questions.
> > > > >
> > > > > Best regards,
> > > > >
> > > > > Authors.

---

### Official Review · Reviewer_6mXX · 2023-11-07

**Soundness:** 3 good
**Presentation:** 3 good
**Contribution:** 3 good
**Rating:** 5
**Confidence:** 3

**Summary:**

The paper motivates the necessity of fact-verification specifically in an unsupervised way. Given that the recent works have focused on NLIs, this work focuses on a pre-training objective that includes claim-fact distillation loss, intra-sample contrastive loss, and scoring loss. These losses are determined based on the positive and negative samples and their embeddings from the knowledge-base conditioned on the claim. The overall goal is to pre-train this model in the context of available knowledge base to verify facts. The results show that the model performs well on FEVER dataset.

Concerns of this work:
1. Size of the models: Given the aspect of large language models where the sizes are in billions, the evaluation is performed on smaller models. Is there some conclusion that can be made with these smaller models instead of just mentioning that the results are good?
2. Datasets: The work has been evaluated only on one single dataset which begs the question of generalizability. Some works such as [1, 2] have evaluated on other datasets such as UKP and FEVER 2 etc. How does this work compare to those? This is particularly necessary because of the use of Wikidata5m knowledge-base that is used. If it is outside the context of Wikipedia, how can this approach work for other knowledge-bases?
3. Comparisons to other approaches: The top-k fact retrieval seems to play an important role, given that the recall in number of facts is improved based on the ranking your approach has, is it a fair comparison to other approaches that work on probably the only retrieved fact? If K=1 then the dev numbers are not comparable to any of the approaches mentioned in the paper.
4. Self-supervision: There is a strong assumption that there is availability of a knowledge base - Wikidata5m and since Fever is derived from it, the losses are carried between facts from Wikidata and Fever claims. Would be good to clarify why the authors think this is self-supervised?


[1]: Incorporating External Knowledge for Evidence-based Fact Verification
[2]: Retrieval-augmented generation for knowledge-intensive nlp tasks

**Strengths:**

1. The paper describes a novel approach for fact-verification
2. Results show significant gains in comparison to state of the art approaches

**Weaknesses:**

1. Generalizability of the approach given other knowledge bases
2. Self-supervision is an ambitious claim
3. Fever is the only dataset used

**Questions:**

In the Summary

---

> ### Author Response · Authors · 2023-11-17
> **Responding to Reviewer 6mXX**
>
> Thank you for your thoughtful and detailed comments on our work. We have provided a high-level overview of major additions in the general rebuttal and hope to address more targeted concerns in this comment.
>
> ## Experiments on Additional Datasets
>
> As this is a common question between some reviewers, we have provided experiments on additional datasets in the general rebuttal. In particular, we benchmark SFAVEL's unsupervised pre-trained model in the FB15k-237 dataset, which is another common benchmark for fact checking tasks. When compared against state-of-the-art methods, our approach is able to achieve +5.3%, +3.6% and +6.3% for Hits@1, Hits@3 and Hits@10 for fact ranking, respectively.
>
> ## Experiments on Larger Language Models
>
> As this is a common question between some reviewers, we have provided experiments on larger language models in the general rebuttal. In summary, we have added experiments with GPT-2-XL (1.5B parameters) as the distilled language model and updated the paper accordingly.
>
> ## Clarification on Comparisons to Other Approaches
>
> The use of top-k retrieval in our approach is motivated by the need to consider multiple pieces of evidence when verifying a claim in real-world applications, and we acknowledge this is a critical factor in our (and all recent) fact verification approach. Given the complexity of some of the claims in the benchmarked datasets (FEVER, FB15k-237), it is not possible to only use single-fact reasoning to predict whether a claim is true or false. For this reason, all state-of-the-art approaches we compare against have a (multiple) evidence retrieval stage (see e.g. [1, 2, 3] for some examples). Therefore, we would like to kindly emphasise that with respect to the top-k retrieval mechanism, the comparison of our method against others is fair and consistent with the literature.
>
> ## Clarification on Self-supervision
>
> We would like to clarify that the main contribution of our work is the unsupervised (self-supervised) pre-training framework. The design of our method is motivated by the goal of learning high-quality claim-fact embeddings useful for downstream fact verification tasks. The embeddings obtained with SFAVEL can be used as input for training any downstream model on different tasks related to fact checking. This is a standard practice in self-supervised learning (see e.g. [4, 5]).
>
> In our experiments we use a linear probe, which is just a linear projection layer, as an example of a downstream model using SFAVEL’s claim embedding and facts pooled embeddings as the input for claim classification. This is required for the case of FEVER, as the task is to classify claims, but it is not for FB15k-237, where only ranking of claims given a fact is important. On the latter, we do not need a supervised stage, and only use the ranking given by SFAVEL directly for evaluation. The aim of this experiment is two-fold: first, we show the generalizability of our unsupervised pre-training method on fact verification tasks with different nature, and second we obtain state-of-the-art performance with a simple linear layer classifier relying on pre-trained SFAVEL features, showing the quality of the learned embeddings. We have now clarified this with the following changes: we added “Pretraining” in the paper title, clarified the optionality of the downstream task in Figure 1, and detailed the purpose for the Linear Probe in the description in Section 4.1.
>
> We appreciate your feedback and will consider these points in our discussions and future work.
>
> Please let us know if you would like to see additional experiments, and thank you again for helping us improve this work.
>
> ## References
>
> [1] Krishna et al., "ProoFVer: Natural Logic Theorem Proving for Fact Verification". TACL, 2021.
>
> [2] Zhong et al., “Reasoning Over Semantic-Level Graph for Fact Checking”. ACL, 2020.
>
> [3] Chen et al., “GERE: Generative Evidence Retrieval for Fact Verification”. SIGIR, 2022.
>
> [4] Bouniot et al., “Proposal-Contrastive Pretraining for Object Detection from Fewer Data”. ICLR, 2023.
>
> [5] Hamilton et al., “Unsupervised Semantic Segmentation by Distilling Feature Correspondences”. ICLR, 2022.

---

> > ### Author Response · Authors · 2023-11-20
> > **A gentle reminder**
> >
> > Dear Reviewer 6mXX,
> >
> > We sincerely appreciate your time and effort in reviewing our paper. We believe that your comments would further strengthen our paper. During the discussion period, we have made every effort to faithfully address all your comments in the responses and revision. Therefore, we we would appreciate if you could read our responses and reconsider your rating on our work.
> >
> > We thank you again for your time and efforts in reviewing our paper.
> >
> > Please let us know if you have any further questions.
> >
> > Best regards,
> > Authors.

---

> > > ### Comment · Area_Chair_GvDp · 2023-12-03
> > >
> > > Dear reviewer 6mXX. Please kindly respond to the authors and let us know if you'd changed your scores. Thanks.

---

### Official Review · Reviewer_1Hnq · 2023-11-08

**Soundness:** 4 excellent
**Presentation:** 4 excellent
**Contribution:** 3 good
**Rating:** 8
**Confidence:** 3

**Summary:**

The authors propose a new (contrastive) loss to train models for unsupervised fact verification. This allows to check claims without having to collect annotations, instead relying on unsupervised claim-fact alignment.

**Strengths:**

* The paper is very well written and well motivated.
* The results presented in the paper are impressive, outperforming FEVER SOTA even for supervised approaches.
* The authors compare the approach on 7 different models, including a variety of small to medium size models.
* The paper contains good ablation experiments, in particular analysing the different components of the loss on a small model.

**Weaknesses:**

* No large models were included, the biggest model tested has 250M parameters. There is no strict definition of LLM, but the authors may overpromise in their title/intro when no model with more than 1B parameters is included.
* The increase over the SOTA may be exaggerated, given that most of the systems the paper compares to are several years old, and do not include the latest generation of models. (This is not strictly a weakness, but context worth mentioning.)

**Questions:**

* Have you considered including larger language models (given that the title mentions "Large Language Models")?

---

> ### Author Response · Authors · 2023-11-17
> **Responding to Reviewer 1Hnq**
>
> Firstly we would like to thank the reviewer for their thoughtful comments and support of our work. We have provided a high-level overview of major additions in the general rebuttal and would like to address more targeted questions in this comment.
>
> Regarding Large Language Models (LLMs), we would like to point out that we do not state our method is targeting LLMs in particular, but more generally any Language Model (LM). However, we agree that assessing the performance of our unsupervised pre-training methodology by distilling from even larger models is an interesting experiment given the current landscape in terms of language model sizes.
>
> Therefore, we have added an additional experiment in Table 2, where we utilise GPT-2-XL as the teacher/distilled language model. Despite not existing a strict definition of LLM, any language model with at least 1B parameters is usually deemed as “large”, and GPT-2-XL contains 1.5B parameters, which effectively makes it a LLM. To run the experiment, we utilise the same hyperparameter settings as the initial experiments.
>
> As a result of this new experiment, we see a 90.32% label accuracy in the FEVER dev dataset. From this experiment we can conclude that our distillation method is still beneficial on larger models, and still provides performance improvements on the downstream task.
>
> Finally, we added additional experiments on the FB15k-237 dataset to showcase the applicability of our method in a different setting.
>
> Thank you again for the helpful feedback and support of our work!

---

> > ### Comment · Reviewer_1Hnq · 2023-11-23
> > **Reply to authors**
> >
> > Thank you for addressing the questions, and I appreciate the new experiments.
> >
> > I enjoyed reading the paper, and still think it's good -- so my recommendation stays the same at accept.

---

> > > ### Author Response · Authors · 2023-11-23
> > > **Thank you**
> > >
> > > Dear Reviewer 1Hnq,
> > >
> > > We really appreciate the time you spent reviewing our work and going through our rebuttal, and are very thankful for your support of our work.
> > >
> > > Best regards,
> > >
> > > Authors.

---

### Official Review · Reviewer_bggr · 2023-11-10

**Soundness:** 3 good
**Presentation:** 3 good
**Contribution:** 2 fair
**Rating:** 5
**Confidence:** 5

**Summary:**

This paper focused on unsupervised fact verification --- verifying a claim based on a trustworthy knowledge base without a requirement of direct supervision. The author proposed to train a scoring model on top of embeddings produced by a pre-trained language model to determine whether a given claim can be aligned to a fact from knowledgebase. The scoring model is trained by leveraging positive and negative examples constructed based on triples from a knowledge graph. The author conducted experimental evaluations on FEVER, and showed that their method can yield a significant improvement over SOTA.

**Strengths:**

1. The author's idea of leveraging a knowledge graph to produce positive and negative examples of unlabeled claims to train a scoring model is creative.
2. The paper is very well-structured, and easy to follow.
3. The experiments presented promising results on FEVER (~8% improvement on accuracy), and the method can work on a broad set of language models.

**Weaknesses:**

1. The technique proposed in the paper does not seem to be generalizable. Specifically, the positive and negative examples constructed through triples from knowledge graph are too simple, which makes this method difficult to generalize to more complicated claims. Specifically, triples can only represent who did what, while in reality, a claim can be who did what at where on when for why.  Any wrong information about these factors can make a claim false. While it is not clear to me how the current method can learn a model that can be effectively aware of some more fine-grained factual differences.
2. The experimental setup is limited. The evaluations are only based on FEVER, which is not convincing. FEVER is created through Wikipedia, and Wikipedia information is closer to triples, which is bias to author's method and training process. At least, an experiment to show the effectiveness of this method on other fact verification dataset would be very helpful.
3. Ranking may not be the best problem formulation for fact verification. For claim verification, it is important to help people decide whether they should believe the claim or not. Now the author formulates this problem as a ranking problem, which is not very useful from a fact verification perspective. it is not clear what does it mean to the user that a claim can find a piece of evidence with 0.9 score.

**Questions:**

1. How this method is different from a ranking/retrieval problem? Is fact verification equivalent to ranking/retrieval?
2. How would this method work on other datasets that are not created based on Wikipedia?

---

> ### Author Response · Authors · 2023-11-17
> **Responding to Reviewer bggr**
>
> We appreciate your thoughtful and detailed comments on our work. We have provided a high-level overview of major concerns in the general rebuttal and address more targeted questions in this comment.
>
> ## Experiments on Additional Datasets
>
> As this is a common question between some reviewers, we have provided experiments on additional datasets in the general rebuttal. In particular, we benchmark SFAVEL's unsupervised pre-trained model in the FB15k-237 dataset, which is another common benchmark for fact checking tasks. When compared against state-of-the-art methods, our approach is able to achieve +5.3%, +3.6% and +6.3% for Hits@1, Hits@3 and Hits@10 for fact ranking, respectively.
>
> ## Clarification on Generation of Negative Instances
>
> We understand your concern regarding the generalizability of our proposed negative examples construction technique and its simplicity. As you correctly pointed out, the individual triples used in our experiments capture relatively simple relationships, typically involving subject-predicate-object structures. While it is true that these triples might not encompass all the complexity of natural language claims, the effectiveness of such negative triplet generation procedures has been demonstrated in previous works (see e.g. [1, 2, 3], to mention a few). In this regard, one of our main contributions is to demonstrate the potential of leveraging structured knowledge from knowledge graphs to improve unsupervised pre-training for fact verification. Despite the fact that real-world claims can involve various aspects such as "who did what, where, when, and why", we argue that breaking down these statements into unitary components can be an effective strategy allowing to capture fine-grained factual differences. As an example, for the claim “Bob Ross created ABC drama The Joy of Painting” (see Section A.1 of the Appendix), our model is capable of obtaining a set of facts that when used together allows the model to perform multi-level reasoning.
>
> ## Relation Between Ranking/Retrieval and Fact Verification
>
> Your comment raises a valid concern about the choice of problem formulation, specifically the use of “ranking” in the context of fact verification, which may not appear intuitive in this context at a first glance. However, ranking is a crucial intermediate step in state-of-the-art fact verification machine learning approaches (see [1] for a comprehensive overview of relevance of ranking in fact verification). While our approach does provide a score for each fact, such as 0.9, it is intended as a measure of the relevance and credibility of the evidence. Regarding the difference between retrieval and ranking, the former aims to find relevant data points (i.e. facts), while ranking goes further by ordering them based on relevance or quality. In the context of fact verification, ranking helps users prioritise evidence, which is crucial when dealing with a large volume of potentially relevant information, as is the case for the benchmarked datasets. Your feedback has been instrumental in highlighting this aspect, and we will consider it for future improvements to our work.
>
> Please let us know if you would like to see additional experiments, and once again, thank you for the helpful and constructive feedback to improve our work!
>
> ## References
>
> [1] Guo et al., “A Survey on Automated Fact-Checking”. TACL, 2022.
>
> [2] Jiang et al., “Don’t Mess with Mister-in-Between: Improved Negative Search for Knowledge Graph Completion”. EMNLP, 2023.
>
> [3] Komatani et al., “Knowledge Graph Completion-based Question Selection for Acquiring Domain Knowledge through Dialogues”. IUI, 2021.
>
> [4] Zhang et al., “Simple and automated negative sampling for knowledge graph embedding”. VLDB, 2021.

---

> > ### Author Response · Authors · 2023-11-20
> > **A gentle reminder**
> >
> > Dear Reviewer bggr,
> >
> > We sincerely appreciate your time and effort in reviewing our paper. We believe that your comments would further strengthen our paper. During the discussion period, we have made every effort to faithfully address all your comments in the responses and revision. Therefore, we we would appreciate if you could read our responses and reconsider your rating on our work.
> >
> > We thank you again for your time and efforts in reviewing our paper.
> >
> > Please let us know if you have any further questions.
> >
> > Best regards,
> > Authors.

---

> > > ### Comment · Area_Chair_GvDp · 2023-12-03
> > >
> > > Dear reviewer bggr. Please kindly respond to the authors and let us know if you'd changed your scores. Thanks.

---

> ### Comment · Reviewer_bggr · 2023-12-03
> **Thanks for the response.**
>
> I appreciated the author's response and the experiments. However, I still have concerns about using FB15k-237. First, this is not a fact-checking dataset, why can not use real fact checking claims from websites such as snopes, factcheck.org etc., Second, this is still a dataset consisting of triples, which has a big overlap with Wikipedia, which may be biased to author's method, similar to FEVER. Therefore, I still have concerns about the generalizability of the method on fact-checking.

---

### Author Response · Authors · 2023-11-17
**General Rebuttal**

Firstly we would like to thank all of the reviewers for their detailed and thoughtful comments. In this post, we summarise our replies to address common concerns, and we are providing more targeted responses to comments raised by reviewers as a reply to their corresponding posts. We also have updated the manuscript to address the necessary changes.

## Experiments on Additional Datasets (bggr, 6mXX, GMHM)

We have added a new experiment on a second dataset for fact checking, FB15k-237, in Section A.2 of the Appendix. In this experiment, we compare the ability of our pre-training technique to provide high-quality claim-fact alignments for ranking facts, as calculated by the Hits@k metric. In this task, we obtain the top K facts given a claim directly from SFAVEL’s unsupervised pre-trained model, without the need for a downstream prediction head. When compared to five recent state-of-the-art methods [1, 2, 3, 4, 5], our model is able to achieve +5.3%, +3.6% and +6.3% for Hits@1, Hits@3 and Hits@10, respectively. This additional experiment shows the benefit of using the SFAVEL's embeddings and rankings, learned entirely in an unsupervised manner, for a variety of downstream fact checking tasks.

## Experiments on Large Language Models (1Hnq, 6mXX)

We have added an additional experiment in Table 2, where we utilise GPT-2-XL as the teacher/distilled language model. Despite not existing a strict definition of "Large" Language Model (LLM), any language model with at least 1B parameters is usually deemed as “large”, and GPT-2-XL contains 1.5B parameters, which effectively makes it a LLM. To run the experiment, we utilise the same hyperparameter settings as the initial experiments. As a result of this new experiment, we see 90.32% label accuracy in the FEVER dev dataset. From this experiment we can conclude that our distillation method is still beneficial on larger models, and still provides performance improvements on the downstream task. We have updated the paper results in Section 3 accordingly.

## Additional Experiments on Loss Function Ablation Study (GMHM, kcpN)

We have updated Table 3 to include the loss function ablation experiments on the best performing backbone (GPT-2-XL) as well as a different, smaller backbone (T5-Small). This allows for a comprehensive understanding of the most critical components of the loss function across two different backbone settings. We consistently observed that the distillation loss is the most relevant component of the loss function, and the contrastive and scoring loss components allow for further improvement in the quality of the learned embeddings.

## Corrected Typos and improved clarity (kcpN, bggr, 6mXX, GMHM)

At the request of reviewer kcpN, we have made a pass through the entire paper and corrected notation typos in Section 3, made explanations more concise, and clarified how the intra-sample contrastive loss solves the cold start problem. Also, at the request of reviewers 6mXX and GMHM, we have added more emphasis on what we believe is the main contribution of our paper: the pre-training framework. To do so, we have clarified this in the abstract and introduction, and suggest adding the word “Pretraining” to the title: Unsupervised $\textbf{Pretraining}$ for Fact Verification by Language Model Distillation. Finally, we have clarified further the purpose of the linear probe used for fine-tuning evaluation.

## References

[1] Krishna et al., "ProoFVer: Natural Logic Theorem Proving for Fact Verification". TACL, 2021.

[2] Zhong et al., “Reasoning Over Semantic-Level Graph for Fact Checking”. ACL, 2020.

[3] Chen et al., “GERE: Generative Evidence Retrieval for Fact Verification”. SIGIR, 2022.

[4] Bouniot et al., “Proposal-Contrastive Pretraining for Object Detection from Fewer Data”. ICLR, 2023.

[5] Hamilton et al., “Unsupervised Semantic Segmentation by Distilling Feature Correspondences”. ICLR, 2022.

---

### Meta-Review · Area_Chair_GvDp · 2023-12-10

**Metareview:**

The paper presents an unsupervised fact verification approach, SFAVEL, focusing on claim alignment with a knowledge base. The proposed method involves training a scoring model on embeddings generated by a pre-trained language model, leveraging positive and negative examples from a knowledge graph. Experimental results on the FEVER dataset demonstrate a significant improvement over the state-of-the-art.

The strengths lie in the innovative use of a knowledge graph for positive and negative examples. Reviewers generally appreciate the paper's clear presentation, motivation, and impressive results outperforming FEVER's state-of-the-art.

Weaknesses:
1. Method's generalizability due to overly simplistic examples from the knowledge graph and limited dataset diversity.
2. Concerns about the absence of large models and the context of comparing to older models temper the overall contribution.
3. The paper's emphasis on ranking as opposed to claim verification's practical utility.
4. The experimental setup's focus on FEVER, derived from Wikipedia, introduces bias, and the lack of comparison to other datasets raises questions about generalizability.

**Justification For Why Not Higher Score:**

While the paper makes a valuable contribution to unsupervised fact verification, concerns about generalizability, dataset bias, and the clarity of certain aspects need to be improved.

**Justification For Why Not Lower Score:**

All weaknesses raised by reviewers seem to have been addressed to some extent by the authors' rebuttal, and several reviewers raised their scores from rejection to acceptance. I think this paper has its value to be published for the community.

---

### Decision · Program_Chairs · 2024-01-16

Accept (poster)